# Development of A Spatiotemporal Database for Evolution Analysis of the Moscow Backbone Power Grid

**Andrey Karpachevskiy \*** , **German Titov** and **Oksana Filippova**

Department of Cartography and Geoinformatics, Faculty of Geography, Lomonosov Moscow State University, 119234 Moscow, Russia; gherman.s.titov@gmail.com (G.T.); o.g.filippova@infraeconomy.com (O.F.)
\* Correspondence: karpach-am@yandex.ru

**Abstract:** Currently in the field of transport geography, the spatial evolution of electrical networks remain globally understudied. Publicly available data sources, including remote sensing data, have made it possible to collect spatial data on electrical networks, but at the same time a suitable data structure for storing them has not been defined. The main purpose of this study was the collection and structuring of spatiotemporal data on electric networks with the possibility of their further processing and analysis. To collect data, we used publicly available remote sensing and geoinformation systems, archival schemes and maps, as well as other documents related to the Moscow power grid. Additionally, we developed a web service for data publication and visualization. We conducted a small morphological analysis of the evolution of the network to show the possibilities of working with the database using a Python script. For example, we found that the portion of new lines has been declining since 1950s and in the 2010s the portion of partial reconstruction reached its maximum. Thus, the developed data structure and the database itself provide ample opportunities for the analysis and interpretation of the spatiotemporal development of electric networks. This can be used as a basis to study other territories. The main results of the study are published on the web service where the user can interactively choose a year and two forms of power lines representation to visualize on a map.

**Keywords:** geographical networks; network morphology; power lines; transport geography; web service

## 1. Introduction

The evolution of road [1–3] and railway networks [4,5], public transport networks [6], transport accessibility [7], and river transport [8] are the study subjects of transport geography [9]. For a long time, this section of geography was dominated by an idiographic approach, which consisted of mostly qualitative studies and simple descriptions of the spatiotemporal features of transport networks [10]. In the middle of the 20th century, the quantitative revolution in geography led to the development of a nomothetic approach. This approach in transport geography involves the use of statistical methods, graph theory, information theory, and other mathematical tools [11]. Network analysis has become one of the main methods in the study of transport systems [12,13]. Researchers collect data using topographic maps, thematic maps and atlases, and statistical materials. Electric networks are a type of geographical network which is currently poorly studied from the point of view of spatiotemporal development. Information about electric networks were classified for a long time due to its strategic importance, as well as due to the tense geopolitical situation of the second half of the 20th century, i.e., the conflict between USA and USSR called the Cold War. Since the beginning of the 21st century, a lot has changed: remote sensing data of ultra-high spatial resolution have become available to a wide range of users, enabling them to quickly collect data on infrastructure networks. Information on electric networks has ceased to be classified, and network companies have begun to publish information on their infrastructure, including schemes of existing and prospective networks.

The analysis of complex networks, developed in the early 2000s, began to be applied to, among other things, electric networks as their classical representative [14,15]. Complex network analysis proposes a method for the evaluation of the electric network as a small-world model or a scale-free model [16,17]. Studies have been dedicated to the vulnerability of electrical networks and the simulations of cascade accidents since it is closely related to the network structure [18–20]. As a study model, researchers often utilize a graph with semantic elements and without geographical references (straight edges connecting nodes) [21,22]; thus, the model becomes greatly simplified, schematized, and deprived of important geographical information. In some cases, the network evolution was studied based on a topological model without taking into account any geographical similarities of the edges to the real-world electric network [23–27]. Such models do not consider the fact that almost every power line is actually made up of different segments constructed in different historical periods due to the peculiarities of the development of electric networks.

The Moscow power system is one of the oldest in the world. It was founded in 1913 with the construction of the 70 kV Bogorodsk–Moscow power line [28], which transmitted electricity produced on local bog peat. The first 220 kV lines (backbone lines by today's standards) were built in 1936. They connected the Stalinogorskaya power plant in the Tula region with Moscow [29]. Later, these lines were complemented with lines from the Uglich and Rybinsk hydroelectric power stations. After World War II, the 220 kV network continued to grow, and in the second half of the 1950s the first 400 (later 500) kV lines from Zhigulevskaya and Volzhskaya hydroelectric power stations were constructed. Due to the long and consistent development of the network, the Moscow power system has become very complex, which makes it interesting to study its spatial evolution over time.

Historically, power grid information has been stored and analyzed using a common information model (CIM). At the same time, it is important to access these data using geoinformation systems (GIS) and utilize its capabilities [30,31]. Power grids belong to the class of spatial networks which support indexing, specific queries, and time-varying properties [32]. Queries to geometry and topological properties can be implemented to spatial networks databases, including extraction of some network indicators (path length, node degree, global efficiency, closeness centrality, and others) [33]. The presence of branches as components of a line single object in such databases can be taken into account through a global unique identifier (GUID) [34]. To develop a spatial network database of a power grid, the following steps are proposed: (1) data acquisition; (2) data preparation and processing; (3) GIS database development; (4) web service development that enables data sharing [35].

There are two basic approaches to the visualization of power grid networks: geographic and topological [36]. Within the topological approach, there exist single line diagrams (electrical connection diagrams) [37,38], time series bar charts/3D surface contours [39], and some others are distinguished. The geographic approach includes the visualization of voltages magnitudes, high voltage line flows, and others [40].

There are studies which suggest visualizing electrical networks within a geographical approach, but with an emphasis on the areal localization of some indicators, for example, power output [41]. The dynamics of power grid networks could be visualized as future state visual representations, including outages and reactive reserve [42], and the multi-scale visualization for planning activities in the power grid [43] or animation loops combining periodic snapshots of the grid into time lapse videos defined across geographic areas [44].

Thus, we see that, despite the rich experience of visualizing various indicators and processes on electric networks in a geographical representation, the issue of database storing and visualizing the historical retrospective of the state of electric networks has not yet been developed. At the same time, the specific spatial development of electrical networks has not yet been studied in research. This study had the following tasks: (1) to collect spatiotemporal data on electric networks; (2) to develop a database structure and a way to represent electric networks; and (3) to develop an algorithm for the automatic morphological classification of networks.

In this study, for the first time, spatiotemporal data on electric networks were collected in such a way as to make it possible to reconstruct the real geometry of the network in detail for each year of its existence from 1936 up to 2020. As a result, we obtained a unique set of network characteristics, which could be very valuable in geographical analysis (including its total length, its morphological structure, and the dynamics of the structure).

It should be noted that, today, there are quite a lot of data sets on electric networks, but none of them are suitable for historical analysis. Thus, the work for this present study resulted in the development of methodology for the collecting, structuring, and performing morphological analysis of spatiotemporal data on backbone electrical networks. The proposed approach for power line visualization could be useful for other specialists in network planning and construction.

## 2. Materials and Methods

### 2.1. Data Collecting

Historically, electric networks were built mosaically, so power lines often consist of segments built in different time periods. This is reflected in the types and forms (also known as species in Russia) of pylons used (a set of pylon forms (species) is called a species composition in Russia [45]), which is very clearly visible both on the ground and in remote sensing data. We prepared a catalog of patterns for image interpretation of pylon species, which provides examples of satellite and ground images of the same pylons (see Supplementary Materials, https://doi.org/10.6084/m9.figshare.16608547.v1 (accessed on 25 November 2021) and https://doi.org/10.6084/m9.figshare.16744354.v1 (accessed on 25 November 2021)). To collect data, we used mainly high-detail images for the period from 2003 to 2020, available in Google Earth [46]. Satellite images allowed us to identify the pylons themselves, their type, and the number of circuits on a pylon (Figure 1). The type of pylon used, in turn, allowed us to estimate in which time period a segment was built. In most cases, we detected the type of pylon using its shadow visible on a satellite image [45,47]. For most of the difficult cases, we conducted ground truthing. Ground truthing was carried out, first of all, for complex parts of the network near large hubs (for example, Chagino, Ochakovo), as well as for some power plants and a number of other difficult cases. We photographed examples of every type of pylon that exists in the Moscow region, which allowed us to create a library of patterns for image interpretation.

For the same purposes, we used declassified images from Keyhole satellites. Keyhole photographic images were downloaded from earthexplorer.usgs.gov [48]. The spatial resolution of photographic images available for the period from 1966 to 1979 allowed us to identify individual pylons. Due to panoramic distortions, the resolution of the image varies from 0.5 m to 2 m. To georeference fragments of these images to the present day satellite image mosaic, we used QGIS [49] (Figure 2). We used a mosaic of modern Google images as a reference image. As a transformation model, we chose a projective model that is best suited for the geometry of the photographic images. Transformation errors and final resolution are shown in Table 1. It should be noted that the spatial resolution was calculated based on scanned image, not on original photographic material. After georeferencing the Keyhole images, we identified the pylons that were vectorized as a point feature class (see Supplementary Materials, https://doi.org/10.6084/m9.figshare.16749199.v1 (accessed on 25 November 2021)).

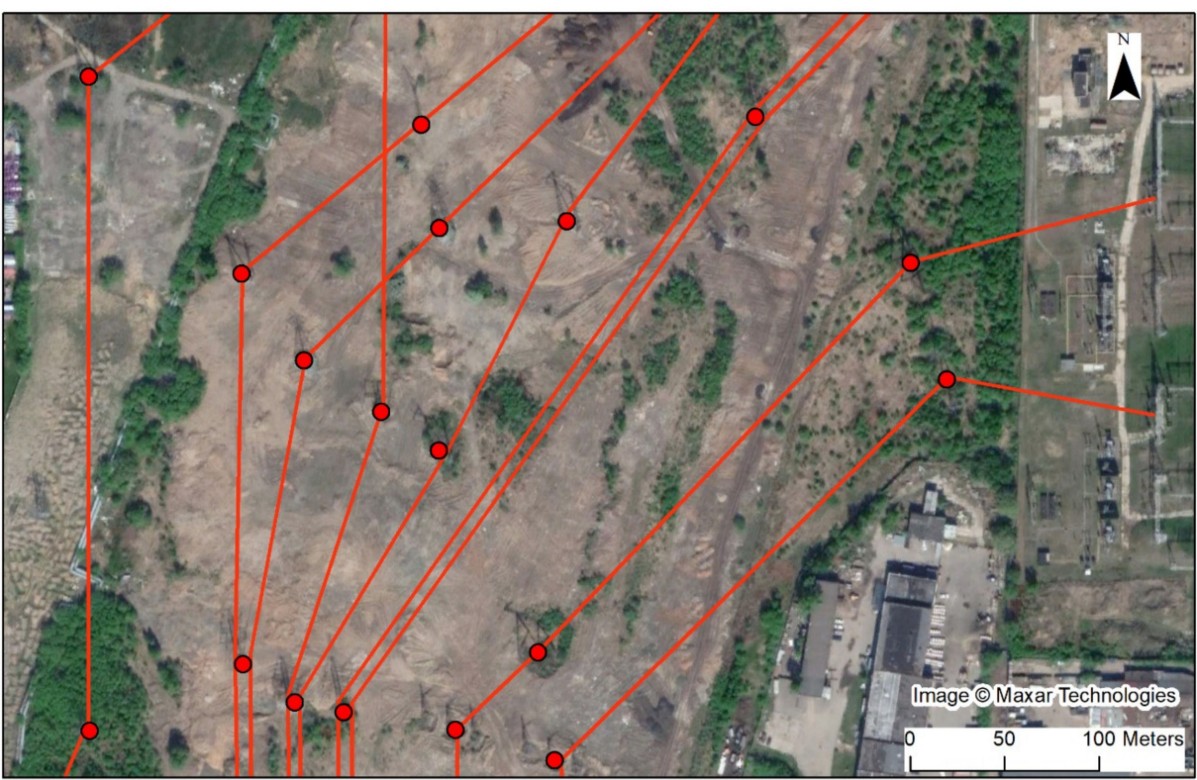

**Figure 1.** Power lines and pylons interpretation using a detailed satellite image in Google Earth (2018).

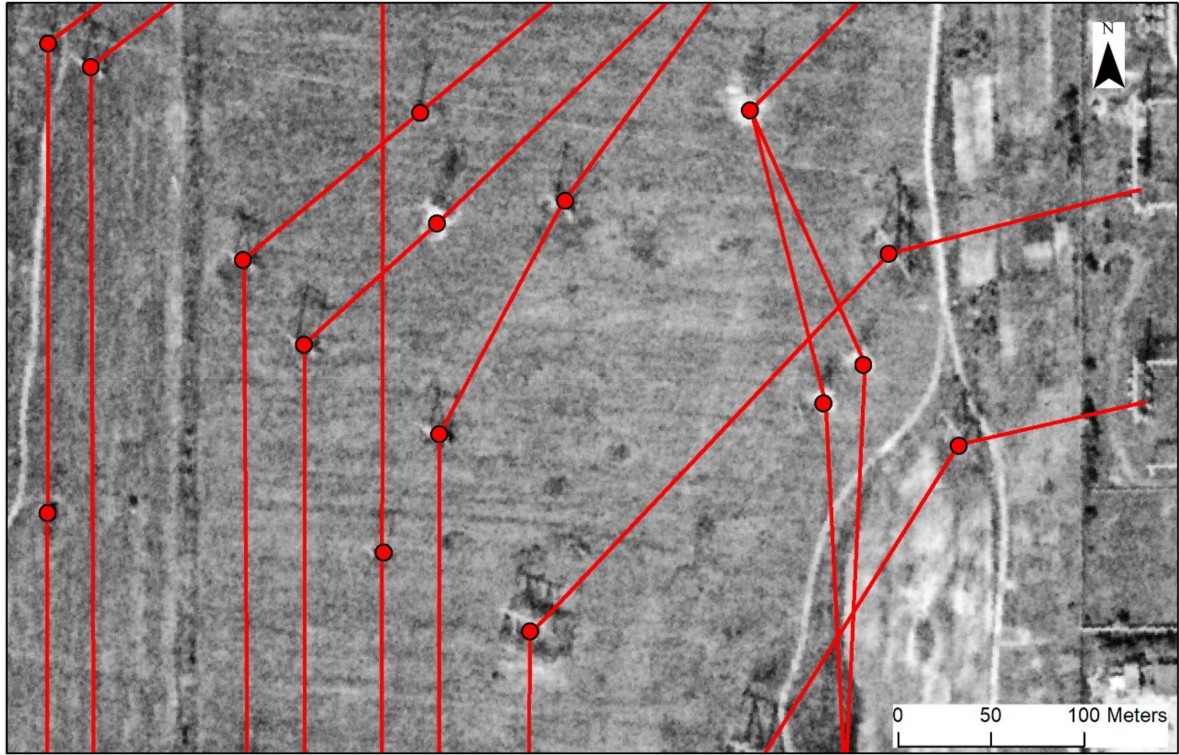

**Figure 2.** Power lines and pylons interpretation using Keyhole series satellite image (1973).

**Table 1.** Characteristics of the used Keyhole images.

| Acquisition Date | Entity ID | Scene Block | Projective Mean Error, m | Scan Resolution, m |
|---|---|---|---|---|
| 1966-07-16 | DZB00403000058H015001 | a, b, c | 0.48 | 0.8 |
| 1966-07-16 | DZB00403000058H014002 | e, f | 5.95 | 0.7 |
| 1972-02-06 | D3C1202-200198A007 | a, b, c | 5.72 | 1.1 |
| 1972-02-28 | D3C1202-400423F011 | d, e, f | 2.24 | 0.8 |
| 1973-09-01 | D3C1206-300465F005 | c, d, e, f | 2.85 | 1.9 |
| 1973-09-01 | D3C1206-300465A007 | f, g | 8.96 | 1.6 |
| 1976-07-11 | D3C1212-100019F010 | c | 6.86 | 1.4 |
| 1976-09-02 | D3C1212-200485A021 | b, c, d, e | 3.5 | 1.4 |
| 1976-09-02 | D3C1212-200485A023 | d, e | 8.96 | 1.6 |
| 1977-07-22 | D3C1213-100136F007 | b, c | 2.3 | 0.9 |

With the use of Google Earth mosaics, available since about 2003, networks were vectorized as linear objects corresponding to individual circuits. The obtained objects were structured in folders and saved as a KML file [50] (a fragment of a KML file is available at https://doi.org/10.6084/m9.figshare.16744405.v1 (accessed on 25 November 2021)). The data obtained from different sources were combined in ArcGIS [51]. The historical state of the networks was modified into a circuit representation (a fragment of the database is available at https://doi.org/10.6084/m9.figshare.16744393.v2 (accessed on 25 November 2021)). For the identification of electrical networks, it is very important to recognize the topomorphological relationships that allow us to correctly identify the joints of different construction segments, correctly understand the network topology, and see the chronological sequence in the network. Figure 3 shows some examples of these relationships. These relationships are essentially a combination of segments dating from different construction periods. Different types and forms of pylons utilized for the same power line often reflect the difference in these segments.

One of the most frequent methods for the connection of some new electrical substations is cutting the nearby existing line along with the construction of new line segments called 'entries'. As a result, two lines appear in the place of the old one. Such a connection scheme, called a 'cut', is also known as the construction of an entry-exit. Branches of the existing lines are quite rarely constructed. As a result, instead of two points, the line can connect three or more using a branch. This scheme is less reliable and often considered to be a temporary solution. The cases of parallel shifting of the lines are, however, rather frequent. These methods avoid two overhead lines crossing each other. Re-routing or the addition of a cable segment is used very frequently in built-up areas.

We used historical schemes available on the website of the Mosenergo History Museum to restore the correct network topology, for example, [52]. For accurate dating, we used the scheme and program for the development of the electric power industry, where the years of commissioning are indicated for lines and substations in a table, for example, [53]. The main issue with this data source is that for each power line, as a rule, only one year is specified. In reality, different segments were commissioned in different time periods due to cuts, re-routing, parallel shifts, and other events.

*2.2. Data Structure*

To organize data within a dataset, we employed tidy data principles [54]. These principles helped to make an easily editable data structure by adding new features. They are also useful for research and visualization using both network analysis tools, web mapping tools, and GIS software.

Power lines and endpoints (such as power stations and substations) are two entities of our dataset. An instance in reference to the power lines is a line segment with constant

characteristics (a single name, a single voltage, a single construction time, etc.). The instance plays the role of an observation point, whereas characteristics act as measured variables.

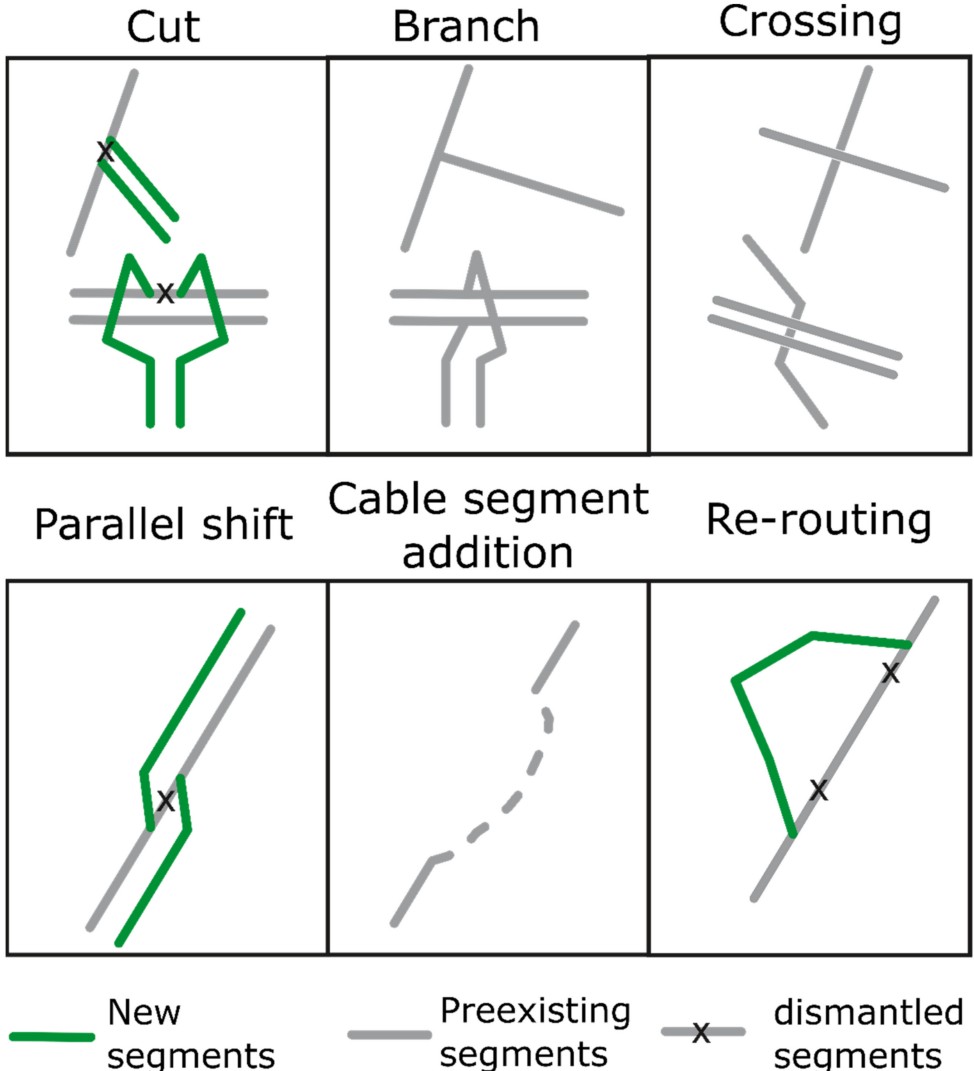

**Figure 3.** Schematic examples of topomorphological relationships.

Each power line has one endpoint at the start and one endpoint at the end. Each endpoint can be a start of one power line or multiple power lines and each endpoint can be an end of one power line or multiple power lines. Geometry is detached from attributes, since equal geometries can represent multiple instances. For example, when a power line changes name or voltage, it becomes a new instance with identical geometry. The detachment helps to eliminate duplicate geometries and reduce data redundancy.

To store the entities and relationships between them, we used the GeoPackage file format. GeoPackage is an open, widely-accepted format for transferring geospatial data, whereas an SQLite database is capable of mapping between tables with the Related Tables extension [55]. The instances are presented in rows and the characteristics are presented in columns. The geometries have a one-to-many relationship with attributes, whereas the endpoints have two one-to-many relationships with power line segments: one for the endpoint at the start and one for the endpoint at the end. Tables 2 and 3 and Figure 4 show the characteristics of power line segments and endpoints.

Table 2. Description of characteristics of power line segments used in the database.

| # | Column Name | Description |
|---|---|---|
| 1 | GID | geometric reference identifier (foreign key) |
| 2 | Year_start | year of construction |
| 3 | Year_end | year of dismantling (where applicable) |
| 4 | geom | line geometry of the power line segment |
| 5 | fid | unique identifier of the instance |
| 6 | Type | type of segment (overhead or underground cable) |
| 7 | Voltage | voltage carried by the lines |
| 8 | Name | name of the instance |
| 9 | Circuit | number of circuit (where applicable) |
| 10 | isBranch | presence of a branch (where applicable) |
| 11 | Branch_points | list of endpoints linked with the branches (where applicable) |
| 12 | Capacity | capacity of instance (where known) |
| 13 | Name_en | name in English |
| 14 | Start_point | start endpoint reference identifier (foreign key) |
| 15 | End_point | end endpoint reference identifier (foreign key) |
| 19 | Year_start_name | start year of instance |
| 20 | Year_end_name | end year of instance |
| 21 | Doubt_years | reference identifier of doubts in correct year value |
| 22 | Doubt_geometry | reference identifier of doubts in correct geometry |

Table 3. Description of characteristics of endpoints used in the database.

| # | Column Name | Description |
|---|---|---|
| 1 | GID | geometric reference (foreign key) |
| 2 | Year_start | year of construction |
| 3 | Year_end | year of dismantling (where applicable) |
| 4 | geom | geometry of the endpoint |
| 5 | fid | unique identifier of the endpoint |
| 6 | Name | name of the endpoint |
| 7 | Number | operation number (where applicable) |
| 8 | Name_en | name in English |
| 9 | Alternative name | alternative name (where applicable) |
| 10 | Class | power station, substation or other |
| 11 | Voltage | highest voltage |

Year_start_name and Year_end_name are borders of the period during which a power line segment has constant characteristics. For example, if the name changed in the year 2000, then this year is the Year_end_name of an old instance and the Year_start_name of a new instance. There are no Year_start_name and Year_end_name in endpoints because they are always equal to Year_start and Year_end. The attribute 'Type' indicates whether this segment is an overhead line or a cable. The attributes 'Doubt_years' and 'Doubt_geometry' are presented as coded values, where 0 indicates no doubts in year value correctness or geometry correctness, respectively, and a value of 1 indicates an inconsistency or lack of initial data.

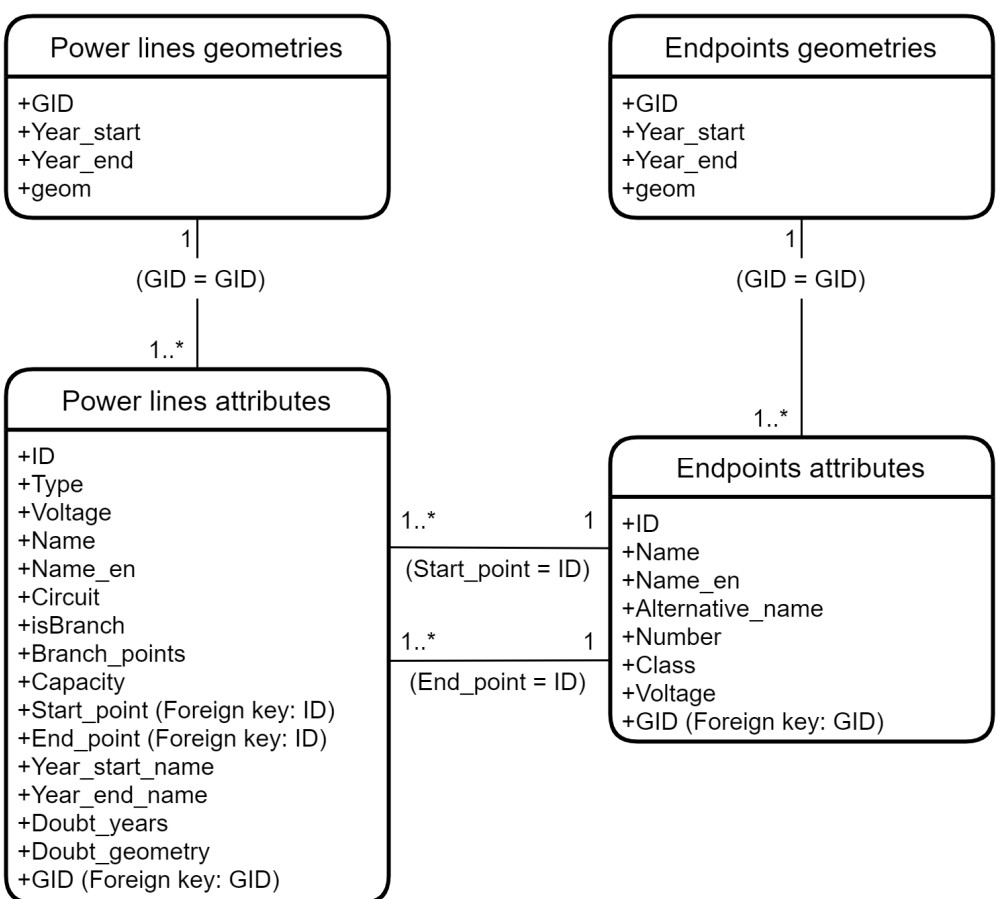

**Figure 4.** Class diagram.

These characteristics for both object classes were divided into two tables corresponding to the geometry of the objects and other characteristics. In addition to the geometry itself, we also included the time frame of the existence of this geometry in the first table ('Year_start', 'Year_end'). This can be explained by the fact that geometry and the time of its existence are inextricably linked characteristics. When the route of a power line changes, the old geometry ceases to exist, and a new one appears. Tables of geometry and other attributes are connected via a key field 'GID'.

The described data structure seems convenient as it helps to store, edit, and query spatiotemporal data in a comfortable way in both GIS and network analysis tools. For example, users can receive information on the state of a power grid for a specified year (e.g., 2000):

SELECT * FROM gPL
JOIN aPL ON gPL.GID = aPL.GID
WHERE aPL.Year_start_name <= 2000 AND (aPL.Year_end_name > 2000 OR aPL.Year_end_name IS NULL)

It is possible to query the length of new lines under construction or dismantled lines for a specific year (e.g., 1958):

SELECT SUM(geometry_Length) FROM gPL
JOIN aPL ON gPL.GID = aPL.GID
WHERE aPL.Year_start = 1958
SELECT SUM(geometry_Length) FROM gPL
JOIN aPL ON gPL.GID = aPL.GID
WHERE gPL.Year_end = 1958

Extraction of the number of cuts and branches for each year is possible using a geoprocessing workflow or scripting. Additionally, we can represent the network development

both by the full cycle construction of completely new lines or by the cuts of the existing lines with construction of entry-exit segments connected to the existing line.

## 3. Results

### 3.1. Morphology Evolution Analysis

To extract data on the state of the network for a specific year, we used queries. Using the script, we classified the changes in the network into construction and dismantling of segments. Each of these segments was also attributed either to the construction of a completely new power line, or to the cutting, branching or re-routing (reconstruction) of the existing power line.

The scripts for morphological analysis of networks were developed using the Python programming language and arcpy library. An approximate scheme of geoprocessing is shown in Figure 5. The script is available online at https://github.com/IOWq750 /historical-power-line-analysis (accessed on 25 November 2021). For analysis, we compared construction and dismantling of adjacent segments. First, we made queries to the database for each year based on the values of the Year_start and Year_end attributes. These queries identified existing segments for each year, as well as those that were constructed or dismantled in that same year. Depending on the status, all segments were assigned the appropriate attribute. After that, the objects were dissolved by the name and status attribute into a composite (multipart) object. The attribute MULTIPART_ID served to count the number of elementary objects included in the multipart object. Then, these multipart objects were converted into single objects, and the dismantling segments turned into two points corresponding to the ends of their segment. Attributes from the new segments, including the names of the starting and ending points, were attached to these points based on spatial proximity. The number of unique item names and their ratios as well as some other attributes indicate the type of modification, which was then used to classify segments.

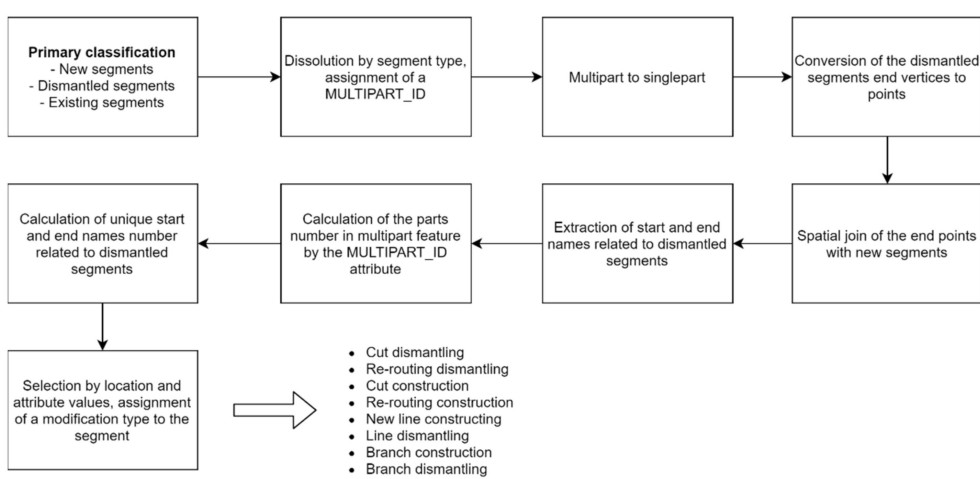

**Figure 5.** Flow chart of morphology types classification.

We summarized the lengths of each segment for each category within one year and five years. The result is represented on the graph in Figure 6; the lengths of segments of each type are shown on the left side, and the total length of the network is shown on the right side. The graph shows the historical periods when new power lines were rarely commissioned due to stagnation, financial crises, and changes in the tariff policy; these are the post-war period of economic recovery, the end of perestroika and the crisis of the 1990s to early 2000s, and the period from 2012 to the present. It is noteworthy how the balance of different types of segments changes over time. Until the end of the 1950s, almost all new segments belonged to new complete lines. Since the 1960s, the share of cut segments has been increasing, as well as the share of extended branches. The second peak



of commissioning of new power lines was observed for the first half of the 1970s. The last peak was observed in 2012, when several lines were being built in the west of the Moscow region in connection with the launch of the Kalininskaya nuclear power plant.

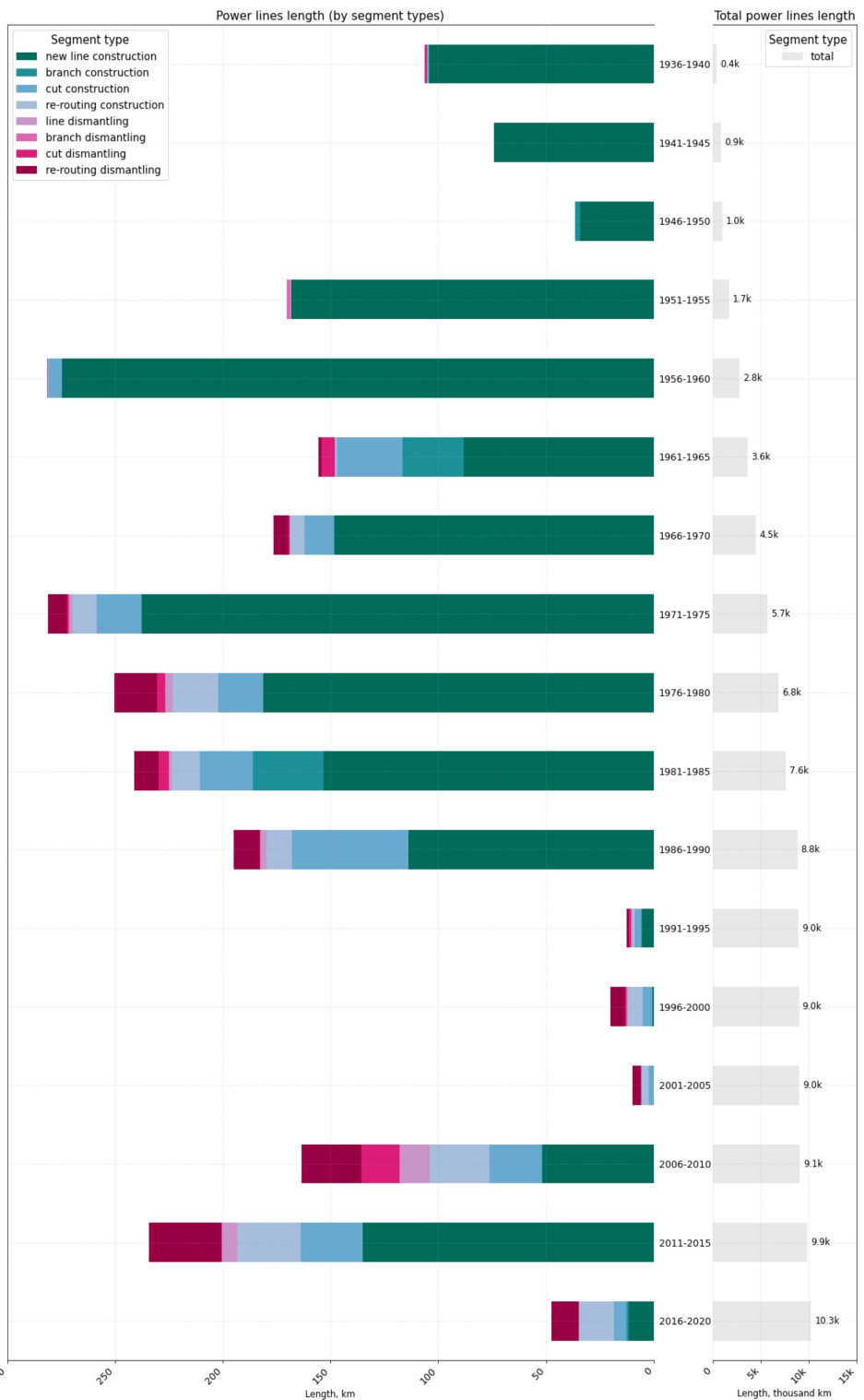

**Figure 6.** Graph of segment lengths categorized by five-year periods, shown in km length. Total line length is given on the right in thousands of kilometers.

Figure 7 shows the ratio of the network modifications of different types. We can see that the first period of the power grid development was dedicated almost exclusively to

the commissioning of new lines. Starting from the second half of the 1960s, systems began to be modified by the cuts of existing lines as well as the construction of new lines. Even in the crisis years of 1990–2005, when no new lines were built at all, the power grid was developing with the cutting existing lines alone.

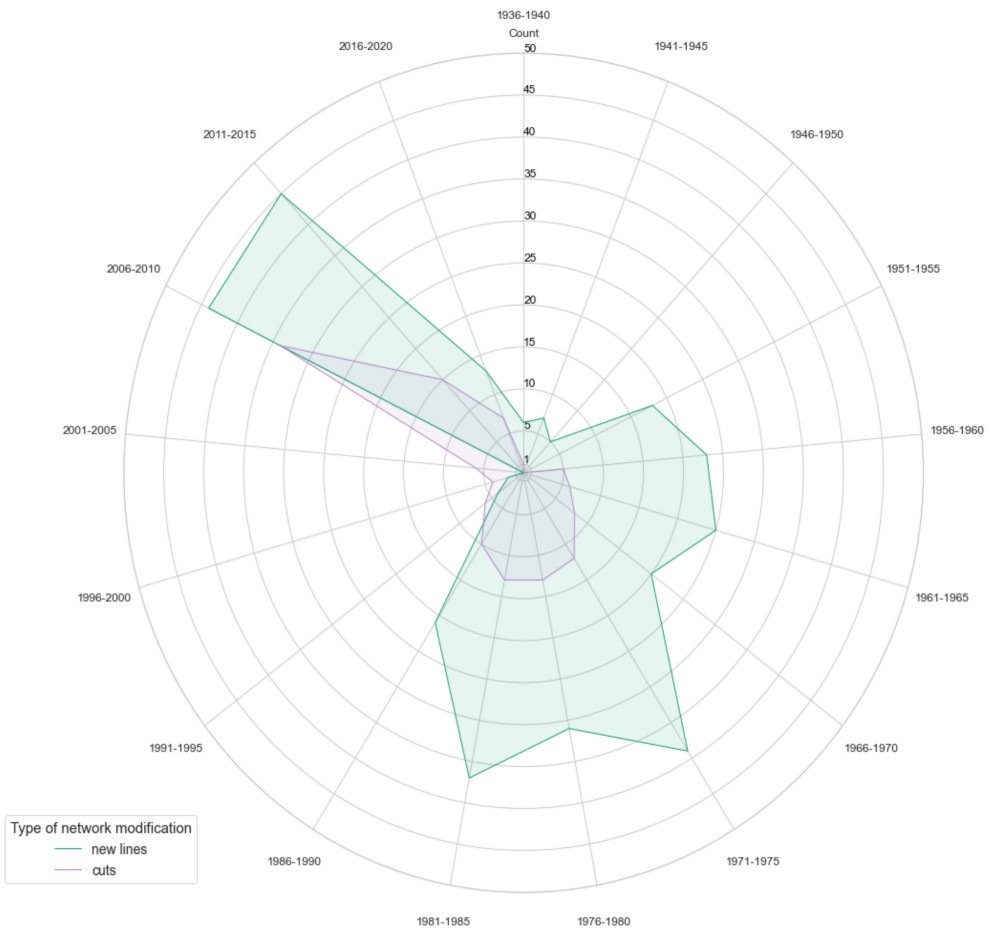

**Figure 7.** Graph of the number of cuts and new lines by five-year period.

### 3.2. Web Service Development

Web map services are useful in communicating spatial data [56,57]. We obtained time slices of the power grid for each year and published them via Web Map Service (WMS) using a QGIS Server and nginx on Ubuntu 18.10. WMS is a standard HTTP interface for requesting geospatial data on the Internet. The WMS capabilities are available at https://powerlines.one/wms?SERVICE=WMS&REQUEST=GetCapabilities (accessed on 25 November 2021).

A sample web map shows WMS with a time filter at the URL https://powerlines.one (accessed on 25 November 2021). The main function of the web map is a WMS viewer. The web map was built using the open-source JavaScript library called 'Leaflet', version 1.7.1. The base layers are the OpenStreetMap's Standard tile layer provided by OpenStreetMap and the Mapbox Satellite layer provided by Mapbox. A custom JavaScript function controls the time filter. A legend of the web map was prepared in advance and rendered as an SVG image overlaid on the webpage. The source code of the web map is available at https://github.com/IOWq750/historical-power-line-analysis (accessed on 25 November 2021).

### 4. Conclusions

The database structure for power line network storing and analysis was proposed. This structure has a spatiotemporal relation that includes geometry and years of object existence and relation with other properties, which could vary during a time period of



object existence. Remote sensing data for the modern period (approximately from 2003 to the present), as well as archival data from declassified satellites of the Keyhole series (the 1970s), allowed us to accurately restore the history of the network in sufficient detail. By interpreting features of individual pylons and their combinations, we identified key morphological features of the network. As part of this study, we collected and structured spatiotemporal data on the backbone electric networks of the Moscow power grid. For the database, we tested an algorithm for morphological classification of modifications in the network, which revealed the geographical features of the system's evolution. The developed data structure turned out to be convenient for morphological analysis, and in the future, we plan to test it on network analysis tools. Even in the crisis years when no new lines were built at all, the power grid was developed and modified by cuts of existing lines. The described approach can also be used for other regional power grids and even for networks with lower voltage classes.

Using WMS with a time dimension helps to share spatiotemporal data in a standard and interoperable way.

**Supplementary Materials:** The following are available online at https://powerlines.one; the satellite image patterns for pylons of 220 kV identification are available online at https://doi.org/10.6084/m9.figshare.16608547.v1; for pylons of 500–750 kV, they are available online at https://doi.org/10.6084/m9.figshare.16744354.v1; a KML fragment with a modern network is available online at https://doi.org/10.6084/m9.figshare.16744405.v1; a database fragment is available online at https://doi.org/10.6084/m9.figshare.16744393.v2; the results of Keyhole imagery interpretation (pylons and some power line segments) are available online at https://doi.org/10.6084/m9.figshare.16749199.v1; our Python script is available online at https://github.com/IOWq750/historical-power-line-analysis.

**Author Contributions:** Conceptualization, A.K. and G.T.; methodology, A.K.; software, A.K., G.T. and O.F.; validation, A.K. and O.F.; formal analysis, A.K.; investigation, A.K.; resources, A.K. and G.T.; data curation, A.K.; writing—original draft preparation, A.K.; writing—review and editing, G.T. and O.F.; visualization, G.T. and O.F.; supervision, A.K.; project administration, A.K.; funding acquisition, A.K. All authors have read and agreed to the published version of the manuscript.

**Funding:** This research was funded by the Council of grants of the President of the Russian Federation, grant number MK-5343.2021.1.5.

**Institutional Review Board Statement:** Not applicable.

**Informed Consent Statement:** Not applicable.

**Data Availability Statement:** The data are not publicly available due to security reasons but they are available for viewing on the site https://powerlines.one (accessed on 25 November 2021).

**Conflicts of Interest:** The authors declare no conflict of interest.

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
