# Peer review of "Development of A Spatiotemporal Database for Evolution Analysis of the Moscow Backbone Power Grid"

_data_

Round 1
Reviewer 1 Report
This paper describes the structure of a spatial database, the data collection procedure and the analysis software implemented for the electricity network of Moscow. The work is interesting, but the presentation needs some improvements.
What is missing is a section about related work. There are some papers about power networks that may be presented and compared with the present paper.
There is a vague point regarding the relationship between “power lines attributes” and “endpoints attributes”. Lines 159-160 mention: “Each power line has exactly two endpoints: one at the start and one at the end. Each endpoint can be a start or an end of multiple power lines”. This phrase describes a many-to-many relationship, not one-to-many, as mentioned (lines 169-170). One the other hand, Figure 5 conceals two one-to-many relationships instead of one: one for the start endpoint and one for the end endpoint. So, in my opinion, figure 5 may depict two one-to-many relationships.
Table 3 does not match with figure 5. The attributes are not the same. In addition, there are attributes that need to be further described:
- Type in class “power lines attributes”
- Class in class “endpoints attributes”
- Doubt_years in class “power lines attributes”
- Doubt_geometry in class “power lines attributes”
It is not clear how the questions mentioned in the following paragraphs may be answered, given the structure of the database. How the attributes help to answer the questions? Sample SQL queries will be useful.
- Lines 187-192: It is possible to query the length of completely new lines under construction for a specific year, extract the number of cuts, branches for each year using a geoprocessing workflow or scripting. Also, we can represent the network development both by the full cycle construction of completely new lines or by the cuts of the existing lines with construction of entry-exit segments connected to the existing line.
- Lines 195-199: To extract data on the state of the network for a specific year, we used queries. Using 195 the script, we classified the changes in the network into construction and dismantling seg- 196 ments. Each of these segments was also attributed either to the construction of a com- 197 pletely new power line, or to the cutting of the existing power line, or to the branch, or to 198 the re-routing (reconstruction) of the existing power line.
Figures:
- In Figure 1, no pylons can be seen.
- There is difficulty in distinguishing something useful in Figure 2 .
- Figure 4 may be omitted, since it gives the same information as figure 5.
Reference [31] is not cited in the text.
The WMS does not work: https://powerlines.one/wms (Service unknown or unsupported)
The paper need to be reviewed by an English language expert. The following phrases are difficult to understand. Consider re-writing them or give more details
- Line 8: including space survey materials
- Line 14: web service for data collection
- Line 25-26: idiographic approach
- Line 28: nomothetic approach
- Line 43-45: This actualized the assessment of the proximity of the electric network to the small-world model or a scale-free model.
- Line 62: HPPs.
- Line 95: We shooted all types of pylons
Author Response
Thank you for your revision! Your comments and remarks helped us improve the manuscript. Below you can read the answers.
What is missing is a section about related work. There are some papers about power networks that may be presented and compared with the present paper.
We have added an experience of power line visualization into the introduction section. At the same time it is very important to understand that in fact there are no studies devoted to historical visualization of power lines to compare with.
There is a vague point regarding the relationship between “power lines attributes” and “endpoints attributes”. Lines 159-160 mention: “Each power line has exactly two endpoints: one at the start and one at the end. Each endpoint can be a start or an end of multiple power lines”. This phrase describes a many-to-many relationship, not one-to-many, as mentioned (lines 169-170). One the other hand, Figure 5 conceals two one-to-many relationships instead of one: one for the start endpoint and one for the end endpoint. So, in my opinion, figure 5 may depict two one-to-many relationships.
We corrected figure 5 and clarified the text.
We replaced “Each power line has exactly two endpoints: one at the start and one at the end. Each endpoint can be a start or an end of multiple power lines.” with ”Each power line has one endpoint at the start and one endpoint at the end. Each endpoint can be a start of one power line or multiple power lines and each endpoint can be an end of one power line or multiple power.”
We replaced “The geometries have a one-to-many relationship with attributes, whereas the endpoints have a one-to-many relationship with power lines.” with “The geometries have a one-to-many relationship with attributes, whereas the endpoints have two one-to-many relationships with power lines: one for the endpoint at the start and one for the endpoint at the end.”
Table 3 does not match with figure 5. The attributes are not the same. In addition, there are attributes that need to be further described:
- Type in class “power lines attributes”
- Class in class “endpoints attributes”
- Doubt_years in class “power lines attributes”
- Doubt_geometry in class “power lines attributes”
We have added the description: Attribute ‘Type’ indicates whether this segment is an overhead line or a cable. Attributes ‘Doubt_years’ and ‘Doubt_geometry’ presented as coded values, where 0 indicates no doubts in year value correctness or geometry correctness respectively, value 1 indicates any inconsistency or lack of initial data.
These characteristics for both object classes were divided in two tables in the database corresponding to geometry of the objects and other characteristics. In addition to the ge-ometry itself, we also included the time frame of the existence of this geometry in the first table (‘Year_start’, ‘Year_end’). This can be explained by the fact that geometry and the time of its existence are inextricably linked characteristics. When the route of a power line changes, the old geometry ceases to exist, and a new one appears. Tables of geometry and other attributes are connected via a key field ‘GID’.
It is not clear how the questions mentioned in the following paragraphs may be answered, given the structure of the database. How the attributes help to answer the questions? Sample SQL queries will be useful.
- Lines 187-192: It is possible to query the length of completely new lines under construction for a specific year, extract the number of cuts, branches for each year using a geoprocessing workflow or scripting. Also, we can represent the network development both by the full cycle construction of completely new lines or by the cuts of the existing lines with construction of entry-exit segments connected to the existing line.
- Lines 195-199: To extract data on the state of the network for a specific year, we used queries. Using the script, we classified the changes in the network into construction and dismantling segments. Each of these segments was also attributed either to the construction of a completely new power line, or to the cutting of the existing power line, or to the branch, or to the re-routing (reconstruction) of the existing power line.
We have added some examples of SQL-queries in text: users can receive information on the state of a power grid for a specified year (e.g. 2000):
SELECT * FROM gPL
JOIN aPL ON gPL.GID=aPL.GID
WHERE aPL.Year_start_name <= 2000 AND (aPL.Year_end_name > 2000 OR aPL.Year_end_name IS NULL)
It is possible to query the length of new lines under construction or dismantled lines for a specific year (e.g. 1958):
SELECT SUM(geometry_Length) FROM gPL
JOIN aPL ON gPL.GID=aPL.GID
WHERE aPL.Year_start = 1958
SELECT SUM(geometry_Length) FROM gPL
JOIN aPL ON gPL.GID=aPL.GID
WHERE gPL.Year_end = 1958
Number of cuts, branches extraction for each year possible using a geoprocessing workflow or scripting - in the next section there is a link at github where you can see a script.
Figures:
- In Figure 1, no pylons can be seen.
Shadows of pylons are clearly visible if you zoom in the figure. Also we have enhanced it by increasing resolution, adding marks for pylons and scale bar.
- There is difficulty in distinguishing something useful in Figure 2 .
In this figure pylons and their shadows are clearly visible and some of them could be matched with pylons from figure 1. We think that it is critical to show readers an example of Keyhole image interpretation so that the methodology would be transparent.
- Figure 4 may be omitted, since it gives the same information as figure 5.
We have removed this figure.
Reference [31] is not cited in the text.
It is cited in line 93. Now it is link 41 in line 118.
The WMS does not work: https://powerlines.one/wms (Service unknown or unsupported)
We have corrected this. The WMS capabilities are available at https://powerlines.one/wms?SERVICE=WMS&REQUEST=GetCapabilities.
The paper need to be reviewed by an English language expert. The following phrases are difficult to understand. Consider re-writing them or give more details
We have revised the manuscript with English language expert.
- Line 8: including space survey materials
We have replaced it with “remote sensing data”
- Line 14: web service for data collection
We have rephrased this part.
- Line 25-26: idiographic approach
Idiographic approach or direction - is a term and there is a link provided at the end of sentence. We have made a tiny rephrase to this part.
- Line 28: nomothetic approach
Nomothetic approach or direction - is a term and there is a link provided at the end of sentence. We have made a tiny rephrase to this part.
- Line 43-45: This actualized the assessment of the proximity of the electric network to the small-world model or a scale-free model.
We have rephrased this part.
- Line 62: HPPs.
We have deciphered the abbreviation.
- Line 95: We shooted all types of pylons
We have rephrased this part.
Reviewer 2 Report
Thank you for inviting me to evaluate the article titled “Development of a spatiotemporal database for evolution analysis of Moscow backbone power grid”.
In my opinion, the research addresses an important and current issue. A new methods and new ways of usage of the GIS and its functionality are useful, and what is more very needed in planning spatial and temporal database. The formal aspects of the paper are proper. The paper is well prepared and well-organized, it brings valuable results however, minor scientific revisions are needed.
The reviewed manuscript focuses on collection and structuring of spatiotemporal data on electric networks with the possibility of their further processing and analysis. The paper is well motivated and it might be relevant to the domain of transport geography science!
I suggest the authors revise the abstract. It would be worthwhile to expand the abstract with the results of the analysis and geovisualisation.
The introduction is well structured, and it covers all the concepts investigated in the methodological part.
I suggest the authors adding more details about the state of the art -- approaches and methods for geovisualization of social or environmental geographic data, other project related to 2D-3D visualization of data related to geovisualization would improve the paper.
Please indicate how your findings can be useful in other engineering disciplines, you can find some related paper in term of spatial and temporal databases:
DOI: 10.1016/j.ssci.2016.09.008
DOI: 10.3390/app11125466
DOI: 10.3390/app10196701
DOI:10.1111/j.1471-0374.2006.00134.x
DOI: 10.3390/land10050492
DOI: 10.36244/ICJ.2020.1.5
The results and conclusions are correctly interpreted, and the discussions are logically related to the outcomes of the research aim. As mentioned before, I consider that this work brings added value in the field and the specific objectives of the manuscript are well related to the previous work developed in this domain.
My questions to the authors (not nessesary built into manuscript):
Can you explain what are the limitations of this method of data collecting in field of GIS?
What processing power do you need to do 3D geovsualization?
Coming to other small observations:
- Figures 1, 2 and10 have a poor quality. The text on these figures is very difficult to read.
- Figures1,2 are missing legend, arrow and scale!
- In some part of Methodology (e.g. software links) is missing references
My evaluation is that the paper is publishable after minor scientific revisions.
Author Response
Thank you for your revision! Your comments and remarks helped us improve the manuscript. Below you can read the answers.
I suggest the authors revise the abstract. It would be worthwhile to expand the abstract with the results of the analysis and geovisualisation.
We have extended the abstract.
I suggest the authors adding more details about the state of the art -- approaches and methods for geovisualization of social or environmental geographic data, other project related to 2D-3D visualization of data related to geovisualization would improve the paper.
We have added an experience of power line visualization into the introduction section. At the same time it is very important to understand that in fact there are no studies devoted to historical visualization of power lines to compare with.
Please indicate how your findings can be useful in other engineering disciplines, you can find some related paper in term of spatial and temporal databases:
DOI: 10.1016/j.ssci.2016.09.008
DOI: 10.3390/app11125466
DOI: 10.3390/app10196701
DOI:10.1111/j.1471-0374.2006.00134.x
DOI: 10.3390/land10050492
DOI: 10.36244/ICJ.2020.1.5
Thank you for these links! We have referenced papers as an example of web service publication:
- Pasquaré Mariotto, F.; Antoniou, V.; Drymoni, K.; Bonali, F.L.; Nomikou, P.; Fallati, L.; Karatzaferis, O.; Vlasopoulos, O. Virtual Geosite Communication through a WebGIS Platform: A Case Study from Santorini Island (Greece). Appl. Sci. 2021, 11, 5466.
- Balla, D.; Zichar, M.; Tóth, R.; Kiss, E.; Karancsi, G.; Mester, T. Geovisualization Techniques of Spatial Environmental Data Using Different Visualization Tools. Appl. Sci. 2020, 10, 6701.
Also, this paper DOI:10.1111/j.1471-0374.2006.00134.x will be useful on the analysis stage later, we think.
My questions to the authors (not nessesary built into manuscript):
Can you explain what are the limitations of this method of data collecting in field of GIS?
Limitations may vary but, in our opinion, the most critical is impossibility of power line topology extraction in some complicated cases, so that we need field observation at best or in the worst case (historical images that represent heavily modified plots) – some archive schemes or maps are necessary.
What processing power do you need to do 3D geovsualization?
At this stage we are collecting data and in future we will move toward deeper analysis and advanced geovisualization methods. Then we may consider 3D visualization as a major option.
Coming to other small observations:
- Figures 1, 2 and10 have a poor quality. The text on these figures is very difficult to read.
We have increased resolution of these images and enhanced contrasting. In fact there is no text, only copyright, so we have enlarged it. We have no figure 10, which did you mean?
- Figures1,2 are missing legend, arrow and scale!
We have added an arrow and scale. There is no need in legend, because only two types of objected are presented in figure, they have different geometry type and description in figure name.
- In some part of Methodology (e.g. software links) is missing references
We have added references.
Round 2
Reviewer 1 Report
Authors may add a section or paragraph about previous related work on spatial network databases in general and power line spatial databases in specific. Some example references follow:
Vassilakopoulos, M. (2009). Spatial Network Databases. In Handbook of Research on Innovations in Database Technologies and Applications: Current and Future Trends (pp. 307-315). IGI Global.
Kanjilal, V., & Schneider, M. (2010). Modeling and Querying Spatial Networks in Databases. J. Multim. Process. Technol., 1(3), 142-159.
Tongyu X., Yuncheng Z., Yingli C. (2012) Topology Analysis and Design of Power Distribution Network Spatial Database Based on GUID Code. In: Wu Y. (eds) Advanced Technology in Teaching - Proceedings of the 2009 3rd International Conference on Teaching and Computational Science (WTCS 2009). Advances in Intelligent and Soft Computing, vol 117. Springer, Berlin, Heidelberg. https://doi.org/10.1007/978-3-642-25437-6_14
Zhang, X., Yuan, X., & Yuan, Y. (2007). New method for designing the spatial database of power network. Electric Power, 40, 75-79.
Yuncheng Zhou, Wei Zheng, T. Xu and L. Fu, "Study on design and maintenance methods of CIM-based spatial database for distribution network," The 2nd International Conference on Information Science and Engineering, 2010, pp. 4062-4066, doi: 10.1109/ICISE.2010.5689832.
Zhang, B. Q., Xiang, Y., Yuan, R. X., & Xu, Z. P. (2006). New method of power network topology analysis based on ComGIS and spatial database. Relay, 34(12), 35-38.
Lin, F., Lin, Y., Zhang, H., & Huang, D. (2021). Power quality monitoring and its visualization application based on graph database. Paper presented at the Proceedings - 2021 6th Asia Conference on Power and Electrical Engineering, ACPEE 2021, 1160-1167. doi:10.1109/ACPEE51499.2021.9436956
Rahman, M. A., Abdul Maulud, K. N., Saiful Bahri, M. A., Hussain, M. S., Ridzuan Oon, A. O., Suhatdi, S., . . . Mohd, F. A. (2020). Development of GIS database for infrastructure management: Power distribution network system. Paper presented at the IOP Conference Series: Earth and Environmental Science, , 540(1) doi:10.1088/1755-1315/540/1/012067
Author Response
Thank you very much for these references!
We have added a paragraph:
Historically, power grid information is stored and analyzed using common information model (CIM). At the same time, it is important to access this data using geoinformation system (GIS) and utilize its capabilities [30, 31]. Power grids belong to the class of spatial networks which support indexing, specific queries and time-varying properties [32]. Queries to geometry and topological properties can be implemented to spatial networks database, including extraction of some network indicators (path length, node degree, global efficiency, closeness centrality and other) [33]. The presence of branches as components of a line single object in such databases can be taken into account through global unique identifier (GUID) [34]. To develop a spatial network database of a power grid following steps are proposed: 1) data acquisition; 2) data preparation and processing; 3) GIS database development; 4) web service development that enables data sharing [35].
Also we have extended the conclusion.
Round 3
Reviewer 1 Report
The comments have been addressed adequately.
This manuscript is a resubmission of an earlier submission. The following is a list of the peer review reports and author responses from that submission.
Round 1
Reviewer 1 Report
The article entitled "Development of spatio-temporal database for evolution analysis of Moscow backbone power grid", presents an incomplete and not well specified methodology and the results of investigating the evolution of the Moscow backbone power grid as a spatio-temporal database, from the construction of the first section until 2020. The methodology focuses on the review of contemporary (21st century) satellite images in google earth on the one hand and Keyhole satellite images from the 1970s on the other hand. Based on the images and especially on the shadows of the pylons used in each period and for each type (voltage) of the network, the authors have been geo-referencing and labelling the sections and their variations by means of attributes, in such a way that by means of some queries on the proposed data model the sections and categories can be recovered or filtered by time slot of a year. The authors have also researched documents that record information on the construction periods of the sections, so that it has been possible to associate an approximate start date with a year's granularity.
The objective pursued is laudable, relevant, provides a better understanding of the evolution of the system and allows for a better understanding of the circumstances or consequences of the electricity transmission system of a region or country.
Criticisms of the document in its present state:
Major:
The introduction lacks, after the motivation and state of the art, the formulation of a need, an issue, a research question or a challenge.
Section 3, which follows section 1 and precedes section 2, is missing.
In section 2, on data and methodology, the section should be strengthened, and the section should be improved so that the methodology can be reproduced, important details are missing. For example, the footprints on the aerial images of the pylons are put as supplementary material. Knowing how many types of pylons, associated to the voltages of the transport network, are necessary and could be included as a table in this section.
It is indicated that in some cases they could not be identified in the images and ground thruthing has been done. It should be indicated how many and what percentage it represents.
When it is indicated that QGis has been used to georeference the Keyhole satellite images, it is not indicated how many images have been used, nor what type of geometric adjustment has been made. There is also no mention of georeferencing residuals. These are factors that affect the accuracy of the generated cartography. Likewise the way of digitising the traces on google earth or google maps. I assume that the georeferencing of the pylons has also been done in QGis using the data model presented on a geopackage file.
When presenting the proposed data model for recording spatio-temporal information, an entity-relationship model has been used. Data models in this domain have long been described in UML as ISO TC211 does for digital geographic information.
It is a lack of internationalisation in the data model as well as in the viewer. The names associated with the lines are in Russian. An additional attribute could have been added for the name with a corresponding English translation.
In the results section, it is indicated that a set of scripted queries are made. It is not indicated what they are, nor in which script language they have been created.
Figure 6 should be improved. On the left hand side the lengths are expressed in km, while on the right hand side (cumulative) in thousands of km. This makes interpretation and comparison or evolution difficult.
In the subsection Web service development, it should be developed a bit more. What kind of tool have you used to create the WMS viewer (Leaflet 1.7), what layers or data sources do you propose as base layers (OSM, MapBox), that the texts of the layer and weather controls are in English and not in Russian.
As for the conclusions section, I do not agree, as it has become clear, with the statement that the methodology is not well described. It is a contribution as the created dataset is not freely available (only at visual level via the WMS).
Nor have they described the algorithm for topomorphological or morphological classification.
Reviewer 2 Report
The manuscript entitled "Development of spatio-temporal database for evolution analysis of Moscow backbone power grid" presents a GIS reconstruction of the power grid system in Moscow through photo-interpretation, integrated by documental analysis.
Unfortunately, this manuscript doesn't satisfy the minimum criteria for being published in a scientific journal: too much information are missing in the methodological part. Too many aspects of the reconstruction of the dataset remain vague or poorly described. And the novelty of this work remains below the average (since it is not true that power grid public datasets do not exist, neither that these datasets are not public...).
Besides: this manuscript has not been checked. It contains some parts of the "template file" in the middle of the main body. Which is unacceptable. Repetitions, grammar mistake and bad logical sentencing is diffuse, creating difficulties in understanding the maning.
You can see my detailed comments in the attached file
